# Research on Hedging Rules Based on Water Supply Priority and Benefit Loss of Water Shortage—A Case Study of Tianjin, China

**Baohui Men [1,\*], Zhijian Wu [1], Huanlong Liu [1], Yangsong Li [1] and Yong Zhao [2]**

[1] Beijing Key Laboratory of Energy Safety and Clean Utilization, North China Electric Power University, Renewable Energy Institute, Beijing 102206, China; 1172211088@ncepu.edu.cn (Z.W.); liuhuanlongHD@163.com (H.L.); lys18811551843@163.com (Y.L.)

[2] State Key Laboratory of Simulation and Regulation of Water Cycle in River Basin, China Institute of Water Resources and Hydropower Research, Beijing 100038, China; iwhrzhy@sohu.com

\* Correspondence: menbh@ncepu.edu.cn; Tel.: +86-010-6177-2451

**Abstract:** When a city's water demand cannot be fully satisfied, the hedging rule can reduce water loss by limiting water supply in advance. Based on water supply priority and benefit loss of water shortage for different users, this paper improved the objective function of hedging rules considering the benefit loss of water shortage. At the same time, according to the idea of restricting water supply by water users in turn, improved hedging rules (IHR) are applied to the urban water supply in Tianjin. The conclusions achieved from this study are as follows: (1) IHR increased water supply assurance rates for domestic water with high-priority and avoided destructive water shortages in agricultural water with low-priority. (2) IHR can better reduce the destructive loss caused by a large number of water shortages and the loss of production caused by a small numbers of water shortages than traditional hedging rules, which ensures high efficiency of water supply during the dry period. The results show that the IHR can improve the operational performance of the urban water supply.

**Keywords:** hedging rules; water supply priority; benefit loss of water shortage; urban water supply

## 1. Introduction

Over the past decades, solving reservoir operation problems has been a challenge for water resource planners and managers. For optimum operations, several rules as well as many simulations and optimization models have been presented, such as real option model, linear decision rule, pack rule, space rule, standard operation policy (SOP), hedging rules (HR) and so on. The SOP is the most common water supply strategy for reservoirs that meet water demand [1]. To fully meet the water demand at present, the SOP saves excess water until abandonment occurs. However, this kind of water supply strategy may cause serious water shortages in the later period. When water shortage occurs, water supply will be used to prioritize the needs of the department with the highest benefits of water use. Therefore, the loss of water shortage is nonlinear with the water shortage. Moreover, the loss of water shortage events with shorter time periods and concentrated water shortages are far larger than a series of water shortage events with long time periods and mild water shortages. To reduce loss of water shortage, Draper and Lund applied the concept of marginal value in economics to the water supply strategy for reservoirs, and proposed to limit water supply when the reservoir has low water supply capacity (storage + inflow) [2]. Hedging rules mean reducing the occurrence of serious water shortages in the future at the expense of small water shortages in the current period [3].

Bower [4] analyzed hedging rules of reservoir scheduling using the theory of system economics for the first time. Shih and Revelle [5] gradually applied hedging rules to reservoir scheduling in the dry

period to reduce the water supply loss. The essence of hedging rules is to save water by frequent small water shortages so as to reduce the risk of severe water shortages in the later period [6]. Hedging rules limit water supply when supply availability is small, the point at which water supply is limited is called the hedging interval [7,8]. The two ends of the hedging interval are called SWA (starting water availability) and EWA (ending water availability) [9], and the ends with larger supply capacity are SWA and EWA [10]. In the hedge interval, the hedge ratio is the amount of water supplied to the amount of supply capacity. Whether the change in hedge ratio is continuous or not, hedging rules can be divided into continuous hedging rules and discontinuous hedging rules [11–14]. There are usually two types of methods for reducing the loss of water shortage by optimizing: (1) Set the SWA and EWA and optimize the hedging ratio; (2) set the hedging ratio and optimize the SWA and EWA. The common objective function in the optimization of hedging rules is the sum of the squared water shortage rate in each period and its deformation. The common forms of several hedging rules are shown in Figure 1, including single-point hedging rules, two-point hedging rules, three-point hedging rules and discrete hedging rules [15–17].

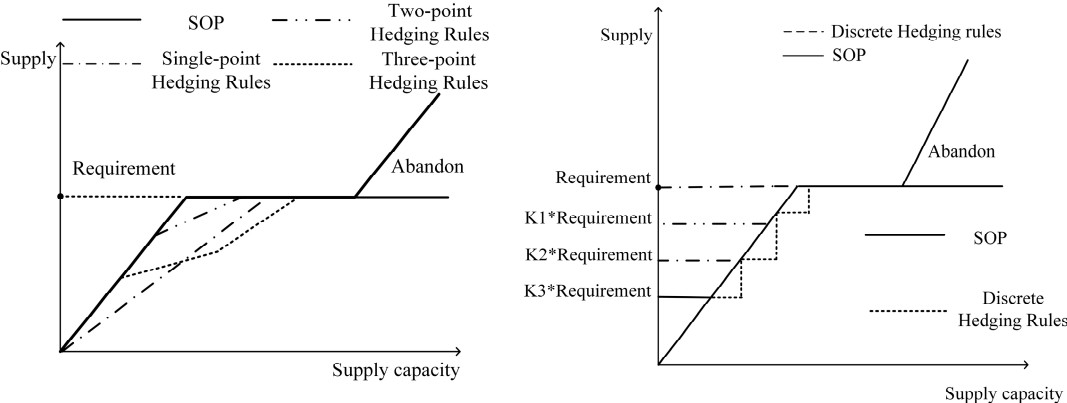

**Figure 1.** Common forms of hedging rules.

Hedging rules can effectively reduce the occurrence of extreme water shortages so they are often used in water supply in areas where water is scarce [18]. Given the form of hedging rules, by giving different objective functions, an optimization algorithm can be used to obtain a better hedging process. In the current study of common hedging rules, the objective function is mostly to reduce the maximum water shortage rate or to reduce the square of the water shortage rate during the study period. Tatano et al. [19] considered the length of water shortage duration in the study of hedging rules from the perspective of water loss and the duration of water shortage. Moy [20] used the hedging rule to minimize the maximum water shortage time. Neelakantan and Pundarikanthan [21] expressed the nonlinear relationship between water loss and water shortage rate by the sum of squared water shortages. Shiau and Lee [22] proposed that the hedging rule has the risk of increasing the total water shortage, and proposed an objective function that can reduce the maximum water shortage and total water shortage during the period. Xu et al. [23] used a risk-averse criterion to rationalize water supply to overcome the shortcomings of the risk-neutral hedging rule in minimizing water shortage impacts in unfavorable realizations, in which actual inflow is less than anticipated. Others enlarged the application scope of hedging theory or improved the hedging process, including limiting the supply of different requirements in turn according to the varied importance of agricultural, industrial and domestic purposes with decreasing supply capacity [24,25].

The above research on urban water supply mostly takes the total water supply and water demand as the research object and does not distinguish between the different types of water users [13,26]. At the same time, the loss of benefits caused by water shortage in different water users during the actual water supply process is neglected. In urban water supply, water users can usually be divided into four categories: Life, industry, agriculture, and ecology [27]. At present, the most common hedging rules are

to study the total water shortage rate, and there are few studies that satisfy the four types of water use. In terms of demand from the four types of water use, life and industrial water demand change little with time, but agricultural and ecological water demand varies greatly with time. Therefore, the water consumption ratio of various types of water demand is different in different time periods. Since the benefits of water supply to different water users are different, in the urban water supply, hedging rules with the squared sum of water shortage rate in each period as an objective function cannot minimize the water loss.

Considering the different water supply benefits and priorities for different water users, in order to minimize loss of water shortage, this paper divides urban water users into four categories (life, industry, agriculture, and ecology), and divides loss of water shortage into two categories: Loss of production caused by minor water shortages and destructive losses caused by severe water shortages. Based on the idea of avoiding destructive losses and minimizing the loss of production and the concept of 10,000-yuan industrial/agricultural value-added water withdrawal, improved hedging rules (IHR) increases the rate of loss reduction rate in the objective function of the hedging rule, fully taking into account the benefits of water supply, so that it can better meet the actual water supply requirements.

## 2. Materials and Methods

### 2.1. The Relationship between Loss and Amount of Water Shortage

There are many water users in urban water supply. According to the different water supply benefits and the importance of water supply, water demand can be divided into domestic water demand, industrial water demand, agricultural water demand and ecological water demand. The benefits of different water supply are different, according to the order of importance of water supply from high to low as follows: Domestic water demand, industrial water demand, agricultural water demand and finally, ecological water demand. When water shortages occur, priority should be given to meeting water users with higher importance. The economic loss caused by water shortages for agriculture and industrial production is called benefit loss of water shortage. In this paper, the water shortage loss is divided into two parts. The shortage of water that causes the reduction of agricultural crops and production in factories is called production loss. Serious water shortages that can cause crops to wither and factories stop production and even close down are called destructive losses. For each water user, the loss of per water shortage (marginal loss of water shortage) increases with the increase of water shortage. The relationship between loss and water shortage can be expressed as shown in Figure 2.

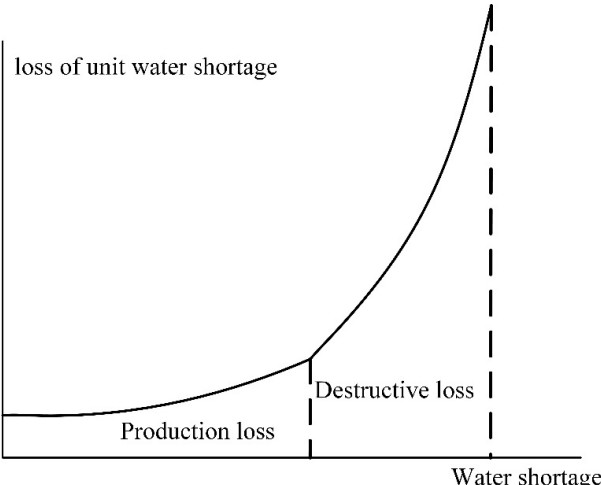

**Figure 2.** Loss of unit water shortage.

The relationship between marginal loss of water shortage and water shortage can be illustrated by the following example. When water shortage is small, agriculture reduces production due to

insufficient supply of water to crops, and industrial production would be affected due to insufficient water supply. At this stage, the marginal loss of water shortage is small, and water shortage would cause less reduction of production [28]. With the increase of water shortage, the crops may appear to wither and the water supply efficiency in the early stage will be lost; thus, industry will stop production due to the large water shortage. These phenomena are all destructive losses of water shortage. When water shortages are more severe, crops may die in agriculture. Since the benefits of agricultural water supply are reflected in the harvest, the death of crops will lead to loss of the benefits of the previous water supply [29]. Therefore, the large water shortage will cause great losses to the crop, which is a destructive loss and should be avoided where possible. For industry, when water shortage is large, it may lead to suspension of production or backlog of raw materials, break of the capital chain, etc., which may lead to the closure of factories. If a factory collapses, the industrial output value will be short-term, which is also destruction loss and should be avoided if possible. For domestic water supply, it plays a vital role in ensuring social stability and is even more important to avoid situations in which water shortage is large. Therefore, when water shortages occur, in order to obtain maximum benefits, agricultural and ecological water supply (with least benefit) will be limited. In order to avoid destructive losses in agriculture when the water shortage is large, the industry water supply should also be limited, and for the same reason, life water supply can also be limited. Based on this idea, when water shortage occurs, according to the order of importance of water supply and the benefit of water supply from high to low, hedging rules take the minimum square of water shortage rate as the objective function can make the water shortage rate process smoother. For the case studied in this paper, it is considered that IHR can avoid destructive losses of water shortage by taking the square of the water shortage rate as the objective function.

### 2.2. Model Establishment and Solving

When water shortage occurs, the water supply should be limited in order of priority from low to high according to the priority of water users. In urban water supply, the priority of water users from high to low is living, industrial, agricultural and ecological water supply. According to this order of priority, this paper developed a ten-day scheduling simulation model based on the available water of local reservoir over ten days. According to the model, the water supply of the reservoir and the water allocation of each user can be determined. The partition of available water of reservoir is shown in Figure 3.

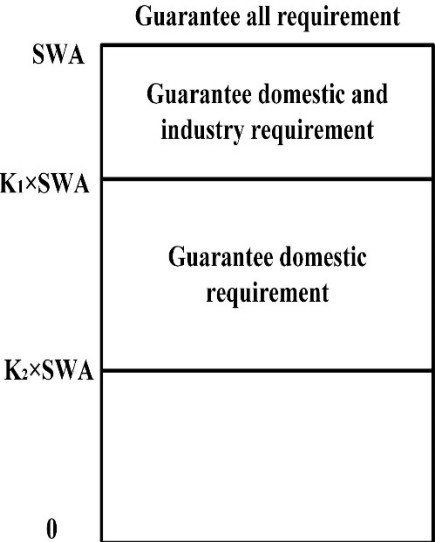

**Figure 3.** Supply capacity partition.

There are many kinds of water sources in urban water supply systems: Surface water, groundwater, and reclaimed water. Some cities also have a desalination water supply. In order to simplify the problem, the reclaimed water, desalinated water, groundwater, and surface water other than the reservoir water are preferentially supplied, and the remaining water is supplied by the local reservoir.

As shown in Figure 3, when available water of reservoir is in guaranteed water supply section, the water is supplied on demand. When available water is in the guaranteed living and industrial section, the water for life and industry is guaranteed, and agricultural water shortage ratio was controlled within 20%. When the available water is in the guaranteed living area, the domestic water is guaranteed, the agricultural and industrial water supply is limited, agricultural water supply equals 80% of demand and industrial water shortage ratio was controlled within 20%. When the available water is below the guaranteed living area, domestic water, industrial water and agricultural water supply equals 95%, 80% and 80% of water demand, respectively. If the available water is too limited to be supplied according to the above water supply, 95% of domestic water demand and 80% of industrial water demand should be met first, followed by the remaining water supply for agriculture and ecology afterwards. The specific process for determining the amount of water supplied and the water allocation of each user is shown as follows:

If $S_k(i) \geq S_w(i)$,

$$D = N_1 + N_2 + N_3 - m_1 - m_2 - m_3 \tag{1}$$

If $k_1 \times S_w(i) \leq S_k(i) < S_w(i)$,

$$
\begin{cases}
l = \frac{S_k(i) - k_1 \times S_w(i)}{(1 - k_1) \times S_w(i)} \\
D(i) = N_1(i) + N_2(i) + l \times N_3(i) - m_1(i) - m_2(i) - m_3(i) \\
d_1(i) = N_1(i) \\
d_2(i) = N_2(i) \\
d_3(i) = l \times N_3(i)
\end{cases}
\tag{2}
$$

If $k_2 \times S_w(i) \leq S_k(i) < k_1 \times S_w(i)$,

$$
\begin{cases}
l = \frac{S_k(i) - k_2 \times S_w(i)}{(k_1 - k_2) \times S_w(i)} \\
D(i) = N_1(i) + l \times N_2(i) + 0.8 \times N_3(i) - m_1(i) - m_2(i) - m_3(i) \\
d_1(i) = N_1(i) \\
d_2(i) = l \times N_2(i) \\
d_3(i) = 0.8 \times N_3(i)
\end{cases}
\tag{3}
$$

If $S_k(i) < k_2 \times S_w(i)$,

$$
\begin{cases}
D(i) = 0.95 \times N_1(i) + 0.8 \times N_2(i) + 0.8 \times N_3(i) - m_1(i) - m_2(i) - m_3(i) \\
d_1(i) = 0.95 \times N_1(i) \\
d_2(i) = 0.8 \times N_2(i) \\
d_3(i) = 0.8 \times N_3(i)
\end{cases}
\tag{4}
$$

where $N_1(i)$, $N_2(i)$ and $N_3(i)$ are water requirement of domestic, industry, agriculture and ecology in $i$th ten days, respectively; $m_1(i)$, $m_2(i)$, $m_3(i)$ are surface water supply, groundwater supply, reclaimed water supply and desalinated water supply in $i$th ten days, respectively; $d_1(i)$, $d_2(i)$, $d_3(i)$ are water supply of domestic, industry, agricultural and ecology in $i$th ten days, respectively; $D(i)$ is the water supplied by reservoir; $S_w(i)$ is supply capacity of the lower limit of guarantee all requirement section; $S_k(i)$ is actual supply capacity in $i$th ten days; and $k_1$ and $k_2$ are pre-set parameters.

In previous research of hedging rules, there are few studies on the profits of different water users; thus, loss of water shortage could not be minimized. The value of industrial and agricultural water supply can be calculated based on the concept of 10,000-yuan industrial added value water withdrawal

and 10,000-yuan agricultural added value water withdrawal, which represent the ratio of industrial (agricultural) water consumption to industrial (agricultural) added value. Usually, the priority of domestic water use is the highest and the priority of ecological water use is the lowest. The value of domestic water supply can be calculated by the higher water consumption of 10,000-yuan industrial value water withdrawal. Similarly, the value of ecological water supply can be calculated by the lower water consumption of 10,000-yuan agricultural value water withdrawal. The target of minimum rate of loss of water shortage is added to the objective function, the objective function is shown as follows:

$$\begin{cases} \text{Min Z} = \sum_{i=1}^{36} \left( \frac{s_i - d_i}{s_i} \right)^2 + k \times \left( 1 - \frac{(n_1 + n_2) \times v_1 + n_3 \times v_2}{(N_1 + N_2) \times v_1 + N_3 \times v_2} \right) \\ d_i = m1_i + m2_i + m3_i + D_i \end{cases} \tag{5}$$

where $i$ represents the ten different days; $s_i$ is the total water demand in $i$th ten days; $D_i$ is the water supply of reservoir in $i$th ten days according to hedging rules; $m1_i$, $m2_i$, $m3_i$ are surface water supply, groundwater supply, reclaimed water supply and desalinated water supply in $i$th ten days, respectively; $d_i$ is the total water supply in $i$th ten days; $N_1$, $N_2$, $N_3$ are total demand of domestic water, industry water, agricultural water and ecology water, respectively; $v_1$, $v_2$ are 10,000 yuan added value water withdrawal of industry and agricultural; $k$ is the weight of the indicator with lowest total water loss rate—the larger $k$ is, the larger the agricultural and ecological water shortages is.

An improved particle swarm optimization algorithm (IPSO) is used for model solving. In IPSO, the sub-individuals are randomly distributed in the feasible domain and when solving, the sub-individual are sorted from optimal to worst according to the function value at first. Then, these sub-individuals are divided into sub-populations, and these subpopulations respectively can be optimized by particle swarm optimization (PSO), respectively. All optimized sub-individuals would form new populations to ensure information sharing. The above process areas performed until the loop termination condition is met and the optimal solution can be reached. In this paper, the above objective function and constraints are used to determine the optimal $k_1$ and $k_2$ values of the reservoir under different water storage conditions, that is, the optimal values of the subgroups. Then, the optimal value of the entire population is obtained by the loop of the optimization algorithm, which is the best $k_1$ and $k_2$. The value of $k_1$ and $k_2$ in this paper are 0.8 and 0.6, respectively.

### 2.3. Research Area

Tianjin is an extremely water-deficient city. In order to solve the water deficiency problem, water from the Yangtze River and Luan River is transferred into Tianjin, and water of the Yellow River can be transferred to Tianjin for emergency use. After the water is transferred to Tianjin, the supply area of transferred water is different. According to transferred water in different areas of Tianjin, Tianjin can be divided into two areas: The water supply area of the Yangtze River and the water supply area of Luan River. There is only local water and water from Luan River in the water supply area of Luan River, while in the water supply area of the Yangtze River there is local water, water of the Yangtze River and water of Yellow River. When water shortages occur, water from Luan River can also be supplied. The water supply network structure of Tianjin is shown in Figure 4. This paper is based on the long series of runoff data from Tianjin in the past 40 years. As hedging rules can play a better role in the case of water shortage, the study selected a typical dry year of 95% frequency of the incoming water, which was 1980. The total water demand is 2.87 billion $m^3$, including 0.615 billion $m^3$ for life, 0.528 billion $m^3$ for industry, 1.197 billion $m^3$ for agriculture and 0.53 billion $m^3$ for ecology. Moreover, the total water supply except the reservoir of Luan River is 2.26 billion cubic meters.

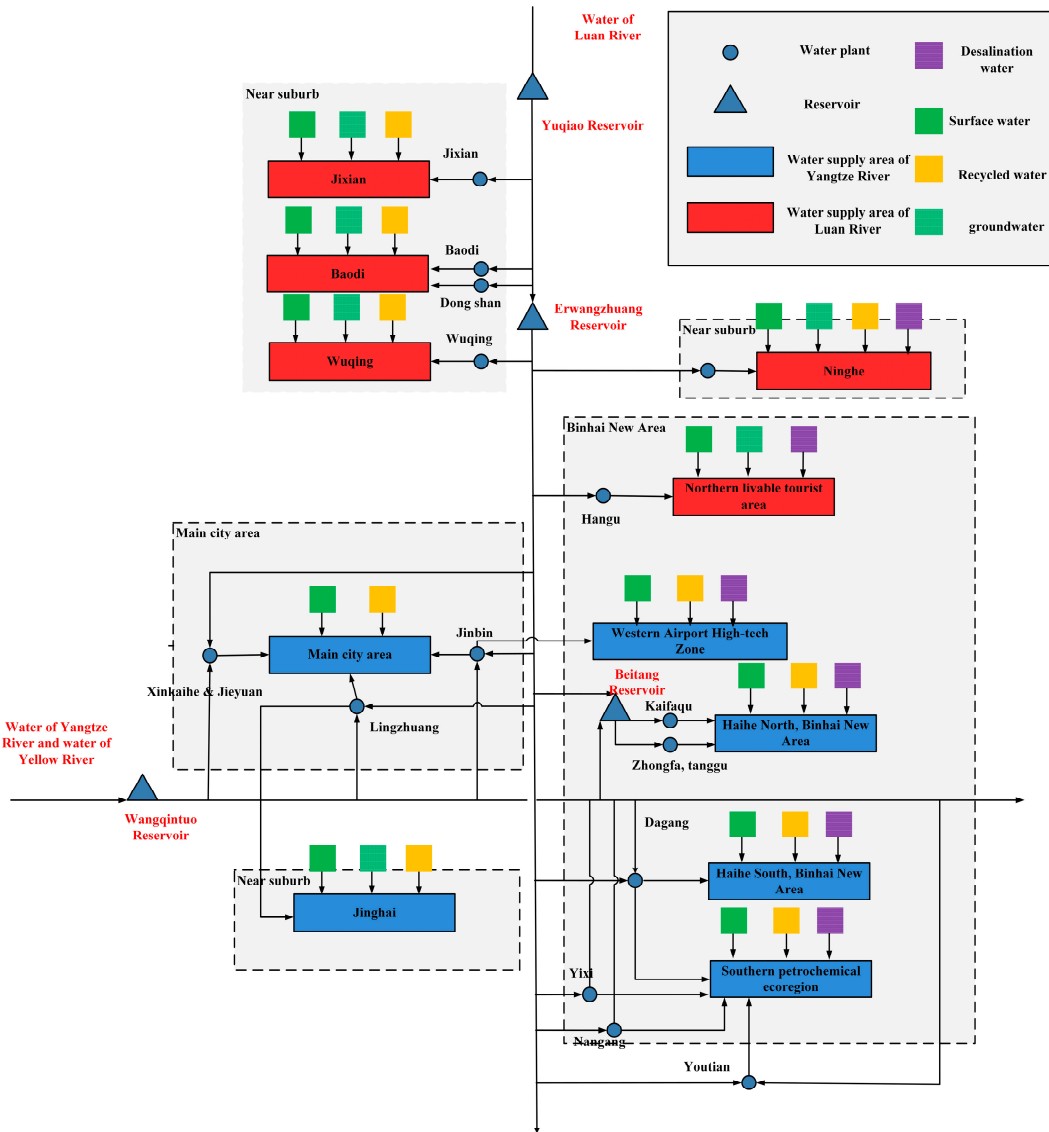

**Figure 4.** Water supply network structure of Tianjin.

## 3. Results

In order to verify the effect of considering the water loss of each water user in the objective function, this paper sets the hedging rule (traditional hedging rule) taking the square of the water shortage rate of urban water supply as the objective function as a comparison. The result obtained by IHR is shown as Scheme 1 and the results of the traditional hedging rules as Scheme 2.

As can be seen from Figure 5, Scheme 2 has less water available than Scheme 1 in only four study series of the whole year. In the actual water supply process, Scheme 2 reserves more water available for future periods. When the water-receiving area and the water-supply area encounter a dry year at the same time, it is important to reserve more water for future periods, which can not only reduce the possibility of destructive water shortage, but also reduce the loss caused by water shortage.

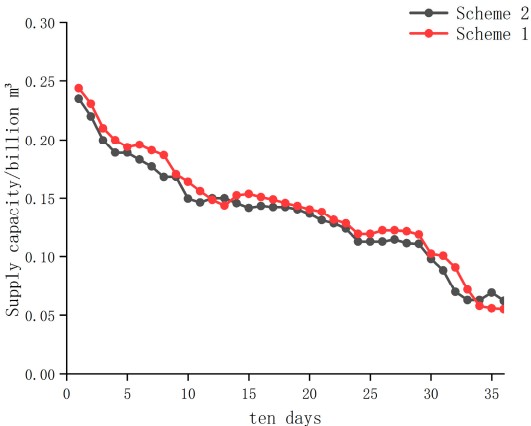

**Figure 5.** Optimization result of Scheme 1 and Scheme 2.

This paper analyzes the results after improving the objective function from the perspective of water users. In order to more clearly express the impact of the improved function's hedging rules on different water users, standard operation rules (SOP) are introduced for comparison. The water shortage in each period of the three water supply schemes (supply water according to IHR, traditional hedging rules and SOP) are shown in the following figures, these three schemes are named Schemes 1, 2, 3. The water shortage results of different schemes are presented in Figures 6–8.

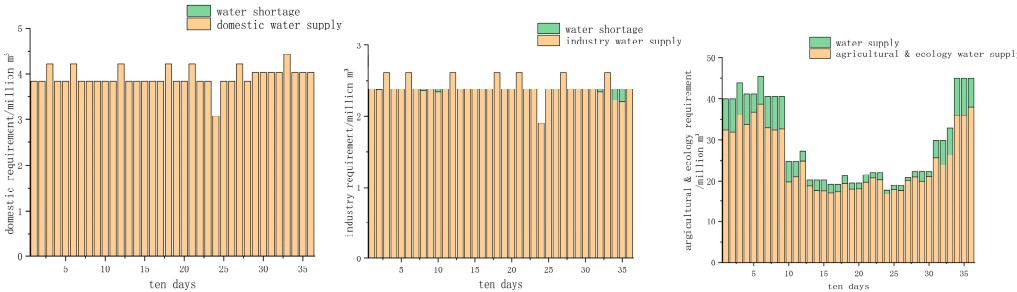

**Figure 6.** Water shortage result of supplying water according to Scheme 1.

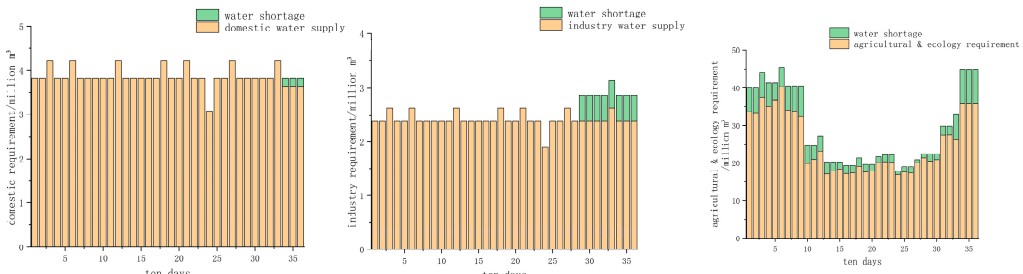

**Figure 7.** Water shortage result of supplying water according to Scheme 2.

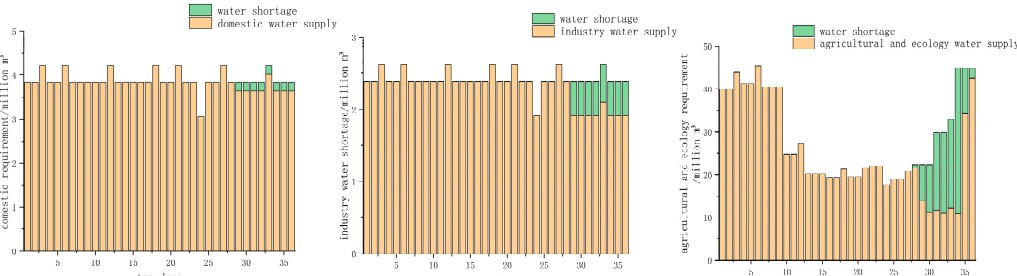

**Figure 8.** Water shortage result of supplying water according to standard operation rules (SOP).

From the perspective of domestic water use, under SOP rules they suffered from water shortage for 8 periods. The traditional hedging rules produced water shortages for 3 periods, and the improved hedging rules of the objective function ensure that the water demand for each period is satisfied. Moreover, the water shortage incident of domestic water occurred at the end of the study period, indicating that the water supply on demand in the early stage under water shortage conditions will lead to water shortage in the future.

When the industrial and agricultural water consumption reaches the maximum value of water shortage, the domestic water supply guarantee rate begins to decrease. Based on SOP, the water shortage period of industrial water occurred four- and six-times more than hedging rules and hedging rules for changing the objective function, respectively, meaning that hedging rules improve the production efficiency brought by industrial water under extreme water shortage conditions. As for agricultural water supply, the hedging rule of changing the objective function has a water shortage event every ten days, indicating that hedging rules ensure the safety of future living and industrial water supply by restricting agricultural and ecological water supply. From the perspective of the degree of damage, the agricultural water supply has been deeply damaged under SOP. The agricultural water shortage from the 30–34 period exceeds the water shortage under the set production loss. In order to ensure domestic water use, it is necessary to drastically reduce agricultural water use, which will undoubtedly cause greater economic losses. It can be seen from the above figures that the use of hedging rules significantly increases water shortage periods of agricultural and ecological water supply, and reduces the water shortage period of living and industrial water supply, which reflects the priority of actual urban water supply. Life, as the most guaranteed water user, needs to be protected as much as possible, and the water supply guarantee rate of agriculture can be sacrificed under the condition of water shortage. The water supply results for the three schemes are shown in Table 1.

**Table 1.** The result of supplying water according to the three schemes.

|  | Scheme 1 | Scheme 2 | SOP |
|---|---|---|---|
| Water shortage (million $m^3$) | 149.54 | 142.91 | 130.69 |
| Loss of water shortage (billion yuan) | 7.758 | 10.58 | 17.115 |
| Period of domestic water shortage happens (maximum water shortage rate) | have no shortage | 34–36 (5%) | 28–36 (5%) |
| Domestic water shortage (million $m^3$) | 0 | 0.58 | 1.75 |
| Period of industry shortage happens (maximum water shortage rate) | 1–36 (8%) | 1–36 (20%) | 28–36 (20%) |
| Industry water shortage (million $m^3$) | 0.46 | 1.6 | 4.34 |
| period of agricultural and ecology water shortage happens (maximum water shortage rate) | 1–14, 15–20, 22–26, 29–36 (20%) | 1–17, 19, 22–24, 30~36 (20%) | 28–36 (76%) |
| Agricultural and ecology water shortage (million $m^3$) | 149.08 | 140.73 | 124.6 |
| Period of water shortage happens (maximum water shortage rate) | 1–36 (18%) | 1–36 (19%) | 28–36 (68%) |

Observing the maximum water shortage rate of four water users, it was found that the maximum water shortage rate of agriculture and ecology was 76% with supply water according to SOP, and the maximum water shortage rate of agriculture and ecology was 20% when water supply was according to Schemes 1 and 2. This proves that the original assumption (hedging rules ensure that water users do not suffer destructive losses by restricting water supply to water users in turn) is reasonable. When calculating the benefits of water supply of Scheme 1, the destructive loss caused by severe water shortage is ignored. The loss of water shortage based on SOP was 17.115 billion yuan, and loss of water shortage would be reduced by 54.6% and 38.2% when supply is according to Scheme 1 and Scheme 2, respectively. When considering the loss of severe water shortage when supplying water according to

SOP rules, hedging rules can further reduce loss of water shortage. In summary, the results proved that hedging rules has a better effect of reducing water loss during the dry period.

Comparing the results of water supply according to Scheme 1 and Scheme 2, the coefficient of variation of the series of water shortage rate is calculated (the coefficient of variation $C_v$ measures the degree of dispersion of series). The coefficient of variation for Scheme 2 and Scheme 1 are 0.46 and 0.48, respectively, and the $C_v$ value for water supply according to the Scheme 2 is smaller, which proved that the water shortage rate process is more gradual. Water supply according to Scheme 1 has a smaller loss of water shortage. Analyzing the cause by observing the water shortage rate process, it was found that the water shortage rate in the early stage of water supply according to Scheme 1 was larger than that of the water supply according to Scheme 2, which increased the volatility of the water shortage rate during the year, but the total water shortage in life and industry during the year was smaller (Scheme 1 does not appear to lack water supply for life). The reason for this is that during the period when agricultural water demand is high in the early stage, Scheme 1 limits the water supply more and saves water to supply the domestic and industrial water in the later stages. Therefore, although the water shortage rate fluctuates slightly, it increases the benefit of water supply (reducing the loss of production).

In summary, the following conclusions can be drawn: During the dry period, hedging rules can reduce destructive losses of severe water shortage (domestic water shortage does not exceed 5%; industrial, agricultural and ecological water shortage does not exceed 20%); hedging rules can reduce the loss of production caused by small water shortages (water loss is smaller); and hedging rules with an improved objective function have a better effect of reducing the loss of production.

## 4. Discussion

Previous research of hedging rules was based on urban water supply reservoirs [30] based on the hedging sub-rules with two triggers for rationing supply (the reservoir operation performance metrics). Both long-term and critical periods are significantly improved relative to the performance metrics of the commonly used single rules based only on the initial storage or water availability [22]. A holistic analysis of the city's water resources yields the principle of total water supply [6]. This paper shows that when water shortage occurs, the difference of water users can improve the rationality of the application of hedging rules. The objective function of hedging rules considering the benefit of water supply is proposed, and the water supply method is applied to the water supply method in Tianjin during the research process. This method of water supply makes the water supply process more uniform, and the small coefficient of variation and the skewness coefficient make the water supply process not have a far-reaching negative impact due to sudden changes in the amount of water in the adjacent period. The IHR reduced the total water shortage in industry and life by 1.14 million m$^3$ and 0.58 million m$^3$, respectively, making hedging rules more realistic. The satisfaction of domestic water use guarantees the basic security of water safety, and the improvement of industrial water supply guarantee rate provides a basis for sustainable social and economic development.

This paper not only uses hedging rules to flatten the process of water shortage, but also divides water shortage losses into two categories: Yield reduction losses and destructive losses. The example of industry and agriculture explains the importance of avoiding the occurrence of destructive losses, and explains that it is more reasonable to divide the water supply by water users in the event of water shortage. On the basis of considering the benefits of different water users, the total water shortage increased the water loss by 2.384 billion yuan compared with conventional hedging rules and avoided damaging losses caused by agricultural shortage under SOP conditions. IHR provides economic protection for the urban water supply security from the economic losses caused by insufficient water resources in the actual water supply process, which provides a theoretical basis for the application of hedging rules in the actual water supply process.

The central idea of IHR is to restrict the water supply to the water users, according to water supply priority and water supply benefit, in order to minimize the water loss when a water shortage occurs.

This study sets the water shortage rate of destructive water shortage loss in agriculture, ecology and industry to be 20%. In fact, different crops, ecological vegetation and industry are affected by water shortages to differing degrees. Therefore, there are two limitations to the improvement of the hedging rules of the objective function: (1) It is difficult to measure the benefits of ecological water supply; (2) the water shortage rate of destructive losses of different water users is not determined based on the impact of water shortage in different industries. In order to break through the above limitations, the future research direction can focus on the following two aspects: According to the ecological development needs of different cities, determine the benefits of ecological water supply; and obtain more reasonable water shortage rate of destructive loss by studying the economic loss caused by the actual water shortage process.

## 5. Conclusions

(1) This paper analyzed and organized the current research of hedging rules, and analyzed the rationality of limiting water supply according to differences between water users when water shortage happens. In the research on reducing loss of water shortage, most current hedging rules summarize the total water shortage rate process by a given objective function, but few studies have analyzed the satisfaction of each water user. This paper divides loss of water shortage into two kinds: Destructive loss of severe water shortage and production loss of small water shortage. Industry and agriculture were used as examples to illustrate the two kinds of losses and explain the importance of avoiding the occurrence of destructive loss of water shortage. Finally, the study explained why it is more reasonable to limit the water supply in turn when water shortage occurs.

(2) IHR was compared with SOP rules and traditional hedging rules without improving objective function. Traditional hedging rules with the objective function of minimum squared sum of water shortage rate can confine the water shortage rate process and reduce the occurrence of destructive loss, but do not perform well in minimizing the loss of production. Therefore, the water supply process is not optimal. IHR improved the objective function when comparing the water supply process and water shortage process with that of hedging rules without improving objective function. It is concluded that after improving the objective function, the destructive loss of severe water shortage and production loss of small water shortage can both be reduced.

**Author Contributions:** B.M. conceived the research theme; Z.W. provided data and designed the analytical approach proposed; H.L. and Y.Z. performed analysis; Y.L. and Z.W. wrote the paper.

**Funding:** This research received no external funding or This research was funded by [the National Key R & D Program of China] grant number [2016YFC0401406] And [the Famous Teachers Cultivation planning for Teaching of North China Electric Power University].

**Conflicts of Interest:** The authors declare no conflict of interest.

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
