# Peer review of "Research on Hedging Rules Based on Water Supply Priority and Benefit Loss of Water Shortage—A Case Study of Tianjin, China"

_water, doi:10.3390/w11040778_

Round 1
Reviewer 1 Report
Article: Research on Hedging Rules Based on Water Supply Priority and Benefit Loss of Water Shortage—a Case Study of Tianjin, China
I agree that it is very important topic. This is a well conducted study, however some of the conclusions/ideas presented here need to be reviewed.
My comments:
Eliminate unnecessary wording through the entire manuscript – for example in conclusion – this paper…this paper ….etc
“The” is overused and there is a lot of passive voice.
-Firstly, please check the structure of the abstract.
-Check the references and add some from international point of view (2018,19..)
-It will be also interesting to point out limitations of the article in one section
- Also to mention some of themes that are important for future research can help to increase the scientific soundness….
Please make introduction shorter and try to reorganize the study according the template.
I like the methodology and figures.
Good luck!
Author Response
Dear Reviewer:
We appreciated very much to Reviewers for your positive comments, the reviewers’ comments and suggestions are very important to improve the manuscript, and the authors thank the reviewers a lot. The relative corrections were listed below.
1. Eliminate unnecessary wording through the entire manuscript – for example in conclusion – this paper…this paper ….etc
Thanks for your reminder. We have improved some of the expressions and eliminated unnecessary wording through the entire manuscript especially in conclusion.
2. “The” is overused and there is a lot of passive voice.
We made some improvements to the expression and deleted some unnecessary words, such as “the”.
3. Firstly, please check the structure of the abstract.
Thanks for your suggestion. We have improved the structure of the abstract.
4. Check the references and add some from international point of view (2018,19..)
Thanks for your reminder. We have added 2 references published in 2018.
Srinivasan, K.; Kumar, K. Multi-Objective Simulation-Optimization Model for Long-term Reservoir Operation using Piecewise Linear Hedging Rule. WATER RESOUR MANAG 2018.
Shiau J. T.; Hung Y. N.; Sie H. E. Effects of Hedging Factors and Fuzziness on Shortage Characteristics During Droughts. WATER RESOUR MANAG 2018, 4, 1-17.
5. It will be also interesting to point out limitations of the article in one section.
The central idea of IHR is to restrict the water supply to the water users, according to the water supply priority and the water supply benefit, in order to minimize the water loss when the water shortage occurs. This study sets the water shortage rate of destructive water shortage loss in agriculture, ecology and industry to be 20%. In fact, different crops, ecological vegetation and industry are affected by water shortage differently. Therefore, there are two limitations to the improvement of the hedging rules of the objective function: (1) it is difficult to measure the benefits of ecological water supply; (2) water shortage rate of destructive losses of different water users is not determined based on the impact of water shortage in different industries. We have added the limitations in line 309-315, page 10.
6. Also to mention some of themes that are important for future research can help to increase the scientific soundness….
In order to break through limitations, the future research direction can focus on the following two aspects: according to the ecological development needs of different cities, determine the benefits of ecological water supply; obtain more reasonable water shortage rate of destructive loss by studying the economic loss caused by the actual water shortage process. We have added the paragraph in line 315-319, page10.
7. Please make introduction shorter and try to reorganize the study according the template.
We have made introduction shorter and deleted some unnecessary representations and have made appropriate improvements to the content of the paper according to the template.
8. I like the methodology and figures.
Thank you for your affirmation and support
Reviewer 2 Report
This article develops an improved hedging rule (IHR) which considers water supply priority and benefit loss of water shortage. The proposed approach is applied to the Tianjin, China and compares with the results of traditional hedging rule and SOP (standard operation rule). This article presents an important issue in water-resources management for urban water supply. However, this article is not well written in model description, data used in research area, and results. This article needs major revisions before it can be considered for acceptance.
1. Unclear meaning of the “benefit loss of water shortage” used in title and there is no clear explanation in text.
2. Page 2, lines 56-58. The authors indicate that “two optimizing methods, one is optimizing hedging ratio, and another one is optimizing SWA and EWA”. Actually, there are studies, such as Shiau (2009) and other studies, optimize SWA, EWA, and hedging ratio simultaneously.
Shiau, J. T., 2009. Optimization of reservoir hedging rules using multi-objective genetic algorithm. Journal of Water Resources Planning and Management. 135(5), 355-363.
3. Page 3, line 98. The “four categories” used in this manuscript should be clearly indicated.
4. Page 3, line 102. Spell out the IHR when it firstly appears in the manuscript.
5. Page 5, lines 160-166. It suggests using consistent description about the water rationing for various water categories when insufficient water supply.
6. Page 5, equations (1)-(4). Sk(i) is not defined.
7. Page 5, lines 178-180. It is unclear what this paragraph means. Besides, what is the relationship between this paragraph and the equations (1)-(4)?
8. Page 5, line 186. How to determine the values of k1 and k2.
9. Page 5, line 189. What is the “10,000-yuan industrial added value water withdrawal” should be briefly described in text.
10. Page 6, equation (5). “m” is not defined.
11. It suggests using consistent symbols in equations (1)-(4) and (5). For example, “(i)” in equations (1)-(4) represents ith ten days, and subscript j in equation (5) represents jth ten days.
12. Page 6, section 2.3 Research area. Brief description about water supply and water demand of the research area is needed.
13. Page 7, Figure 5. The results shown in Figure 5 are based on hydrologic condition of which year should be clearly described. Besides, supply capacity in the vertical axes of this figure representing surface water, groundwater, reclaimed water, reservoir, or total water supply should be clearly indicated in text.
14. Page 8, line 244. What is the “case I”?
15. Page 10, line 307-311. It is unclear what this paragraph associated with references [27]-[29] mean.
16. Some minor editorial suggestions include:
(1) Page 5, line 158. “fig. 4” should be “Fig. 3”
(2) Page 10, line 319. “m3”, “3” should be typed in superscript.
(3) Page 11 lines 366-369, references 3 and 4. Names of authors are not correctly typed.
(4) Page 12, lines 423-424. Reference 29 is identical to reference 16.
Author Response
Dear Reviewer:
We appreciated very much to Reviewers for your positive comments, the reviewers’ comments and suggestions are very important to improve the manuscript, and the authors thank the reviewers a lot. The relative corrections were listed below.
1. Unclear meaning of the “benefit loss of water shortage” used in title and there is no clear explanation in text.
The economic loss caused by water shortage for agriculture and industrial production is called benefit loss of water shortage. In this paper, the water shortage loss is divided into two parts. The shortage of water causes the reduction of agricultural crops and production in factories is called production loss. Serious water shortage can cause crops to wither, factories stop production and even close down, which called destructive water loss. We have added explanation of “benefit loss of water shortage” in line103-107, page 3.
2. Page 2, lines 56-58. The authors indicate that “two optimizing methods, one is optimizing hedging ratio, and another one is optimizing SWA and EWA”. Actually, there are studies, such as Shiau (2009) and other studies, optimize SWA, EWA, and hedging ratio simultaneously.
Shiau, J. T., 2009. Optimization of reservoir hedging rules using multi-objective genetic algorithm. Journal of Water Resources Planning and Management. 135(5), 355-363.
Thank you for your advice, this paper has added your suggested references.
3. Page 3, line 98. The “four categories” used in this manuscript should be clearly indicated.
An explanation has been added to the brackets after “four categories” according to your suggestion.
4. Page 3, line 102. Spell out the IHR when it firstly appears in the manuscript.
We are sorry to forget to spell out the IHR when it firstly appears in the manuscript. The complete words have been added to the place where IHR firstly appeared in the manuscript.
5. Page 5, lines 160-166. It suggests using consistent description about the water rationing for various water categories when insufficient water supply.
Different water users have different water supply priorities. According to the requirements of regional water resources development and utilization, the supply of agricultural water in the dry year is 80% of the required water. There are no provisions for ecological and industrial water supply requirements, so this paper sets the same as agriculture. Domestic water has the highest priority and the required guarantee rate is the highest. The general urban water supply requires a domestic water guarantee rate of over 95%. Therefore, this paper used the consistent description about the water rationing for agriculture, industry and ecology, which was smaller the life.
6. Page 5, equations (1)-(4). Sk(i) is not defined.
We are sorry to forget it. It has been added in Page 5, line 166.
7. Page 5, lines 178-180. It is unclear what this paragraph means. Besides, what is the relationship between this paragraph and the equations (1)-(4)?
We are sorry to forget to delete this paragraph which is in the Word Template. Thank you for your attention.
8. Page 5, line 186. How to determine the values of k1 and k2.
The values of k1 and k2 are obtained by particle swarm optimization algorithm. After the two parameters are determined, the upper and lower limits of the hedge rules (SWA and EWA) are determined.
9. Page 5, line 189. What is the “10,000-yuan industrial added value water withdrawal” should be briefly described in text.
We have added the brief description of “10,000-yuan industrial added value water withdrawal” in line 170, page 5.
10. Page 6, equation (5). “m” is not defined.
The authors selected two typical years in the preliminary study, where “m” represents the different cases of two typical years, and “case I” in the following represents one of the typical years of the preliminary study. However, in this paper, we chose a typical year that best reflects the advantages of IHR, so the “m” in the formula and the description of “case I” in the text need to be deleted.
11. It suggests using consistent symbols in equations (1)-(4) and (5). For example, “(i)” in equations (1)-(4) represents ith ten days, and subscript j in equation (5) represents jth ten days.
Thank you for your suggestion. This paper has corrected the meaning of the same letter in different formulas to be consistent.
12. Page 6, section 2.3 Research area. Brief description about water supply and water demand of the research area is needed.
We have made a brief description about water supply and water demand of the research area in line 199-204, page 6.
13. Page 7, Figure 5. The results shown in Figure 5 are based on hydrologic condition of which year should be clearly described. Besides, supply capacity in the vertical axes of this figure representing surface water, groundwater, reclaimed water, reservoir, or total water supply should be clearly indicated in text.
Thank you for your advice. We have added the description of selected typical year in Line 203-208, page 6. This paper is based on the long series of runoff data from Tianjin in the past 40 years. Because the hedging rules can play a better role in the case of water shortage, the study selected a typical dry year of 95% frequency of the incoming water which was 1980. Supply capacity in the vertical axes of this figure represents reservoir and we have corrected in the figure.
14. Page 8, line 244. What is the “case I”?
The authors selected two typical years in the preliminary study, where “m” represents the different cases of two typical years, and “case I” in the following represents one of the typical years of the preliminary study. However, in this paper, we chose a typical year that best reflects the advantages of IHR, so the “m” in the formula and the description of “case I” in the text need to be deleted.
15. Page 10, line 307-311. It is unclear what this paragraph associated with references [27]-[29] mean.
16. Some minor editorial suggestions include:
(1) Page 5, line 158. “fig. 4” should be “Fig. 3”
Thanks for your reminder, this paper has corrected this error.
(2) Page 10, line 319. “m3”, “3” should be typed in superscript.
Thanks for your reminder, this paper has corrected the same errors.
(3) Page 11 lines 366-369, references 3 and 4. Names of authors are not correctly typed.
Thanks for your reminder, this paper has corrected the same errors.
(4) Page 12, lines 423-424. Reference 29 is identical to reference 16.
We are sorry an thank for your advice. We have corrected the error.
Round 2
Reviewer 1 Report
Thank you for the correstions.
please check:
R. 374 -The authors declare no conflict of interest.
Author Response
Dear Reviewer:
We appreciated very much to Reviewers for your positive comments, the reviewers’ comments and suggestions are very important to improve the manuscript, and the authors thank the reviewers a lot. The relative corrections were listed below.
R. 374 -The authors declare no conflict of interest.
Thank you very much for your reminder. We have added “no” in the sentence.
Reviewer 2 Report
This article develops an improved hedging rule (IHR) which considers water supply priority and benefit loss of water shortage. The proposed approach is applied to the Tianjin, China and compares with the results of traditional hedging rule and SOP (standard operation rule). This revised article makes great improvements according to reviewers’ comments. However, it still needs minor revisions before it can be published.
1. Page 2, lines 56-58. References are needed for various hedging rules (single-point, two-point, three-point hedging rules) listed in this paragraph.
2. Page 5, lines 184-185. The authors indicate that k1=0.8 and k2=0.6 are pre-set parameters. How to determine these two values should be more precisely described in text. In authors’ replies, the authors indicate that “The values of k1 and k2 are obtained by particle swarm optimization algorithm”. How to set the objective function and required constraints in the particle swarm optimization algorithm to obtain the values of k1 and k2 should be clearly described in text.
3. Page 6, lines 221-226. Please check the correctness of the numbers of various demands. Sums of life demand of 0.942, industrial demand of 0.9, agricultural demand of 1.109, and ecology demand of 0.73 are not equal to the total demand of 2.87.
4. Some minor editorial suggestions include:
(1) Page 1, line 43. “Bower et al” should be “Bower”
(2) Page 1, line 44. “Shih” should be “Shih and ReVelle”.
(3) Page 2, line 57. “figure 1” should be “Figure 1”.
(4) Page 2, line 66. “Tatano” should be “Tatano et al”.
(5) Page 2, line 72. “Xu B” should be “Xu”.
(6) Page 6, lines 224-225. “0.942 billion”, “0.9 billion”, “1.109 billion” and “0.73 billion” should be “0.942 billion m3”, “0.9 billion m3”, “1.109 billion m3” and “0.73 billion m3” (3 typed in superscript).
(7) Page 6, line 226. “square meters” should be “cubic meters”.
(8) Page 7, line 238. “figure 5” should be “Figure 5”.
(9) Page 12, lines 447-448, reference #31. Names of authors are not correct.
Author Response
Dear Reviewer:
We appreciated very much to Reviewers for your positive comments, the reviewers’ comments and suggestions are very important to improve the manuscript, and the authors thank the reviewers a lot. The relative corrections were listed below.
1.Page 2, lines 56-58. References are needed for various hedging rules (single-point, two-point, three-point hedging rules) listed in this paragraph.
Thanks for your reminder. We have added 3 reference, including:
Adams, L.E.; Lund, J.R.; Moyle, P.B.; Quinones, R.M.; Herman, J.D.; O'Rear, T.A. Environmental hedging: A theory and method for reconciling reservoir operations for downstream ecology and water supply. WATER RESOUR RES 2017, 53, 7816-7831.
You, J.; Cai, X. Hedging rule for reservoir operations: 1. A theoretical analysis. WATER RESOUR RES 2008, 44.
Tu, M.; Hsu, N.; Tsai, F.T.C.; Yeh, W.W.G. Optimization of hedging rules for reservoir operations. JOURNAL OF WATER RESOURCES PLANNING AND MANAGEMENT-ASCE 2008, 134, 3-13.
2. Page 5, lines 184-185. The authors indicate that k1=0.8 and k2=0.6 are pre-set parameters. How to determine these two values should be more precisely described in text. In authors’ replies, the authors indicate that “The values of k1 and k2 are obtained by particle swarm optimization algorithm”. How to set the objective function and required constraints in the particle swarm optimization algorithm to obtain the values of k1 and k2 should be clearly described in text.
We are sorry for not explaining this problem well before. In this paper, the above objective faction and constraints are used to determine the optimal k1 and k2 values of the reservoir under different water storage conditions, that is, the optimal values of the subgroups. Then the optimal value of the entire population is obtained by the loop of the optimization algorithm, which the best k1 and k2.
3. Page 6, lines 221-226. Please check the correctness of the numbers of various demands. Sums of life demand of 0.942, industrial demand of 0.9, agricultural demand of 1.109, and ecology demand of 0.73 are not equal to the total demand of 2.87.
Because multiple sets of data were used for optimization calculations during the research process, the water demand data of other groups were confused in the manuscript. Now we have rewriten the data as follows: The total water demand is 2.87 billion m3, including 0.615 billion m3 for life, 0.528 billion m3 for industry, 1.197 billion m3 for agriculture and 0.53 billion m3 for ecology.
4. Some minor editorial suggestions include:
(1) Page 1, line 43. “Bower et al” should be “Bower”
Thanks for your advice. We have improved the expression according to your suggestion.
(2) Page 1, line 44. “Shih” should be “Shih and ReVelle”.
OK. We have rewritten this sentence.
(3) Page 2, line 57. “figure 1” should be “Figure 1”.
We are sorry for that. We have corrected the above problem, including “figure 1” and “figure 5”.
(4) Page 2, line 66. “Tatano” should be “Tatano et al”.
Thanks for your advice. We have improved the expression according to your suggestion.
(5) Page 2, line 72. “Xu B” should be “Xu”.
We are sorry for that and corrected the sentence.
(6) Page 6, lines 224-225. “0.942 billion”, “0.9 billion”, “1.109 billion” and “0.73 billion” should be “0.942 billion m3”, “0.9 billion m3”, “1.109 billion m3” and “0.73 billion m3” (3 typed in superscript).
Thanks for your reminder. We have added m3 in the manuscript.
(7) Page 6, line 226. “square meters” should be “cubic meters”.
We are very sorry that something has gone wrong here and the error has been corrected.
(8) Page 7, line 238. “figure 5” should be “Figure 5”.
We are sorry for that. We have corrected the above problem, including “figure 5” and “figure 1”.
(9) Page 12, lines 447-448, reference #31. Names of authors are not correct.
Thank you for your reminder. We have corrected the names of authors.